# YAP determines the cell fate of injured mouse hepatocytes *in vivo*

Norio Miyamura[1], Shoji Hata[1], Tohru Itoh[2], Minoru Tanaka[3], Miki Nishio[4], Michiko Itoh[5], Yoshihiro Ogawa[6], Shuji Terai[7], Isao Sakaida[8], Akira Suzuki[4], Atsushi Miyajima[2] & Hiroshi Nishina[1]

The presence of senescent, transformed or damaged cells can impair tissue function or lead to tumorigenesis; therefore, organisms have evolved quality control mechanisms to eliminate them. Here, we show that YAP activation induced by inactivation of the Hippo pathway specifically in damaged hepatocytes promotes their selective elimination by using *in vivo* mosaic analysis in mouse liver. These damaged hepatocytes migrate into the hepatic sinusoids, undergo apoptosis and are engulfed by Kupffer cells. In contrast, YAP activation in undamaged hepatocytes leads to proliferation. Cellular stresses such as ethanol that damage both liver sinusoidal endothelial cells and hepatocytes switch cell fate from proliferation to migration/apoptosis in the presence of activated YAP. This involves the activation of CDC42 and Rac that regulate cell migration. Thus, we suggest that YAP acts as a stress sensor that induces elimination of injured cells to maintain tissue and organ homeostasis.

[1] Department of Developmental and Regenerative Biology, Medical Research Institute, Tokyo Medical and Dental University (TMDU), M&D tower 21F, 1-5-45 Yushima, Bunkyo-ku, Tokyo 113-8510, Japan. [2] Laboratory of Cell Growth and Differentiation, Institute of Molecular and Cellular Biosciences, The University of Tokyo, 1-1-1 Yayoi, Bunkyo-ku, Tokyo 113-0032, Japan. [3] Department of Regenerative Medicine, Research Institute, National Center for Global Health and Medicine, 1-21-1 Toyama, Shinjuku-ku, Tokyo 162-8655 Japan. [4] Division of Molecular and Cellular Biology, Kobe University Graduate School of Medicine, 7-5-1 Kusunoki-cho, Chuo-ku, Kobe 650-0017, Japan. [5] Department of Organ Network and Metabolism, Graduate School of Medical and Dental Sciences, TMDU, 1-5-45 Yushima, Bunkyo-ku, Tokyo 113-8510, Japan. [6] Department of Molecular Endocrinology and Metabolism, Graduate School of Medical and Dental Sciences, TMDU, AMED-CREST, 1-5-45 Yushima, Bunkyo-ku, Tokyo 113-8510, Japan. [7] Division of Gastroenterology and Hepatology, Graduate School of Medical and Dental Sciences, Niigata University, 757, Ichibancho, Asahimachidori, Chuo-ku, Niigata 951-8510, Japan. [8] Department of Gastroenterology and Hepatology, Graduate School of Medicine, Yamaguchi University, 1-1-1 Minami-Kogushi, Ube 755-8505, Japan. Correspondence and requests for materials should be addressed to H.N. (email: nishina.dbio@mri.tmd.ac.jp).

Cellular stress in tissues and organs leads to senescent, transformed or damaged cells[1–4]. These cells can impair tissue function or lead to tumorigenesis and therefore need to be eliminated and their loss compensated for through cell proliferation to maintain tissue and organ size[5–9]. However, the molecular mechanisms that act to maintain tissue and organ homeostasis during cellular stress are largely unknown.

The liver plays a central role in metabolic homeostasis due to its role in metabolism, and the synthesis, storage and redistribution of nutrients[10,11]. The liver is also one of the main detoxifying organs, removing waste and xenobiotics through metabolic conversion and biliary excretion. The waste and xenobiotics come from the gastrointestinal tract via the portal vein, and diffuse into small blood vessels known as hepatic sinusoids. Thus, the liver is constantly exposed to various stresses. The liver consists of several different cell types including hepatocytes, which have metabolizing and detoxifying abilities, liver sinusoidal endothelial cells (LSECs), which form the sinusoidal wall and cover the hepatocytes, and Kupffer cells, which are sinusoid-resident macrophages.

The Hippo pathway regulates organ size and cancer formation by modulating cell proliferation and death via regulation of YAP activation[12–16]. Central to the Hippo pathway is a kinase cascade wherein Mst (the mammalian orthologue of the *Drosophila* Hippo) phosphorylates and activates the adaptor protein Mob and the protein kinase LATS. Activated LATS then phosphorylates the transcription coactivator YAP, and inhibits its activation by cytoplasmic retention. Unphosphorylated YAP translocates into the nucleus, interacts with the transcription factor TEAD and induces target gene expression. Gene knockout of Hippo pathway components induces hepatomegaly and liver cancer in mice. Recently, we reported that loss of Mob causes YAP activation and cancer formation in mouse liver[17,18]. Depletion of the YAP gene suppressed liver cancer formation in Mob-knockout mice. Thus, the liver phenotypes caused by an impaired Hippo pathway are strongly dependent on YAP.

In this study, we examine the dynamics of YAP-activating hepatocytes by *in vivo* mosaic analysis in mouse and discover that the fate of YAP-expressing hepatocytes changes from proliferation to migration/apoptosis depending on the status (healthy or damaged) of the LSECs.

## Results

**YAP-activated hepatocytes are lost in mouse liver.** To examine how the Hippo pathway affects the fate of individual hepatocytes, we first established mosaic conditions by using hydrodynamic tail vein injection (HTVi) to introduce Myc-tagged YAP-wild type (WT), or one of three active YAP mutants (YAP (1SA), YAP (2SA) or YAP (5SA)), into mouse liver *in vivo*[19,20]. Under such conditions, HTVi achieves exogenous gene expression in ∼30% of hepatocytes (Supplementary Fig. 1a).

Immunofluorescence analysis revealed that YAP (WT) localized in the cytoplasm and active YAP (1SA, 2SA and 5SA mutants), as expected, localized in the nucleus (Fig. 1a, insets). Real-time PCR showed that expression of one of the YAP target genes, connective tissue growth factor (*ctgf*), was increased in YAP (1SA, 2SA and 5SA) but not YAP (WT) during days 1 and 2 (Supplementary Fig. 1b), consistent with functional YAP activation. Interestingly, the number of hepatocytes expressing active YAP markedly decreased during days 3–7 post-HTVi, whereas no such reduction occurred in YAP (WT) livers (Fig. 1a,b). Immunoblotting confirmed a corresponding reduction in the active-YAP but not YAP (WT) protein (Fig. 1c and Supplementary Fig. 1c). To verify that loss of active YAP (5SA) hepatocytes caused this decrease, ROSA26-LacZ reporter mice

were injected with Myc-YAP (5SA)-IRES-Cre plasmid. YAP- and Cre-expressing cells were detected by immunostaining with anti-Myc and anti-β-galactosidase (β-gal) or 5-bromo-4-chloro-3-indolyl-β-D-galactoside (X-gal) staining, respectively. Both cell populations decreased in parallel within 7 days of HTVi (Fig. 1d and Supplementary Fig. 2), confirming the loss of YAP (5SA)$^+$ hepatocytes. Nevertheless, normal liver size was maintained in YAP (5SA) mice (Supplementary Fig. 3a), prompting us to investigate the effect on hepatocyte proliferation. In YAP (5SA), but not YAP (WT), liver at days 3–7 post-HTVi, the proliferation of mainly Myc-negative hepatocytes was increased (Supplementary Fig. 3b–d).

We next investigated whether the above effects could be replicated by activating endogenous YAP via inactivation of the Hippo pathway. We injected LacZ-IRES-Cre plasmid into $Mst1^{\text{flox/flox}};Mst2^{\text{flox/flox}}$ mice and $Mob1a^{\text{flox/flox}};Mob1b^{\text{tr/tr}}$ mice, both of which lack Hippo signalling[17,18,21]. Under such conditions, consistent with activated YAP, *ctgf* expression was upregulated in these mice (Supplementary Fig. 4). Immunofluorescence analysis demonstrated that LacZ-expressing hepatocytes were reduced in both mutant strains within 7 days post-HTVi (Fig. 1e), consistent with our results using exogenous active YAP mutants.

**YAP-activating hepatocytes are engulfed by Kupffer cells.** A previous study reported that hepatocytes expressing activated Ras undergo cellular senescence and are lost by elimination dependent on CD4$^+$ T cells (termed 'senescence surveillance')[22]. To determine whether senescence surveillance also played a role in the loss of YAP (5SA) hepatocytes in our system, we first examined the mouse livers for senescence-associated (SA)-β-gal$^+$ hepatocytes. Forced expression of activated K-Ras (G12V) induced hepatocyte senescence as expected. In contrast, YAP (5SA) hepatocytes were SA-β-gal$^-$ and thus not senescent (Fig. 2a). We also explored more directly whether adaptive immunity was involved in the loss of YAP (5SA) hepatocytes by introducing Myc-tagged YAP (WT)- or YAP (5SA)-expressing plasmids into immunodeficient NOD/Shi-scid, IL-2Rγ-null (NOG) mice by HTVi[23]. Numbers of YAP (5SA) hepatocytes steadily decreased also in NOG livers over 7 days post-HTVi (Fig. 2b). Thus, the elimination of YAP-activated hepatocytes is regulated by a mechanism distinct from senescence surveillance.

To identify this mechanism, we stained mouse liver sections to detect markers of various cell populations. Unlike YAP (WT) hepatocytes, YAP (5SA) hepatocytes migrated to hepatic sinusoids, 'sinusoid intravasation of hepatocytes' (Supplementary Fig. 5a–d) and were surrounded and engulfed by Kupffer cells (liver-resident macrophages) (Fig. 2c,d and Supplementary Movie 1). Of note, we found that Kupffer cells also exist in the immunodeficient NOG mice (Supplementary Fig. 5e). Thus, we depleted Kupffer cells from the YAP-expressing mice using clodronate liposomes[24], and found that the loss of YAP (5SA) cells from the liver was suppressed, and instead TUNEL$^+$ apoptotic cells coexpressing Myc appeared (Fig. 2e,f and Supplementary Fig. 6). These data demonstrate that Kupffer cells play a direct role in the elimination of YAP-activated hepatocytes.

**LSEC and hepatocyte injury switch YAP-activated cell fate.** To determine whether the mosaic ratio affected YAP (5SA) hepatocyte elimination, we introduced a YAP (5SA)-expressing adenovirus vector into mouse liver. This vector induces exogenous gene expression with up to 80% efficiency. We infected mouse livers with sufficient PFU of this vector to induce YAP

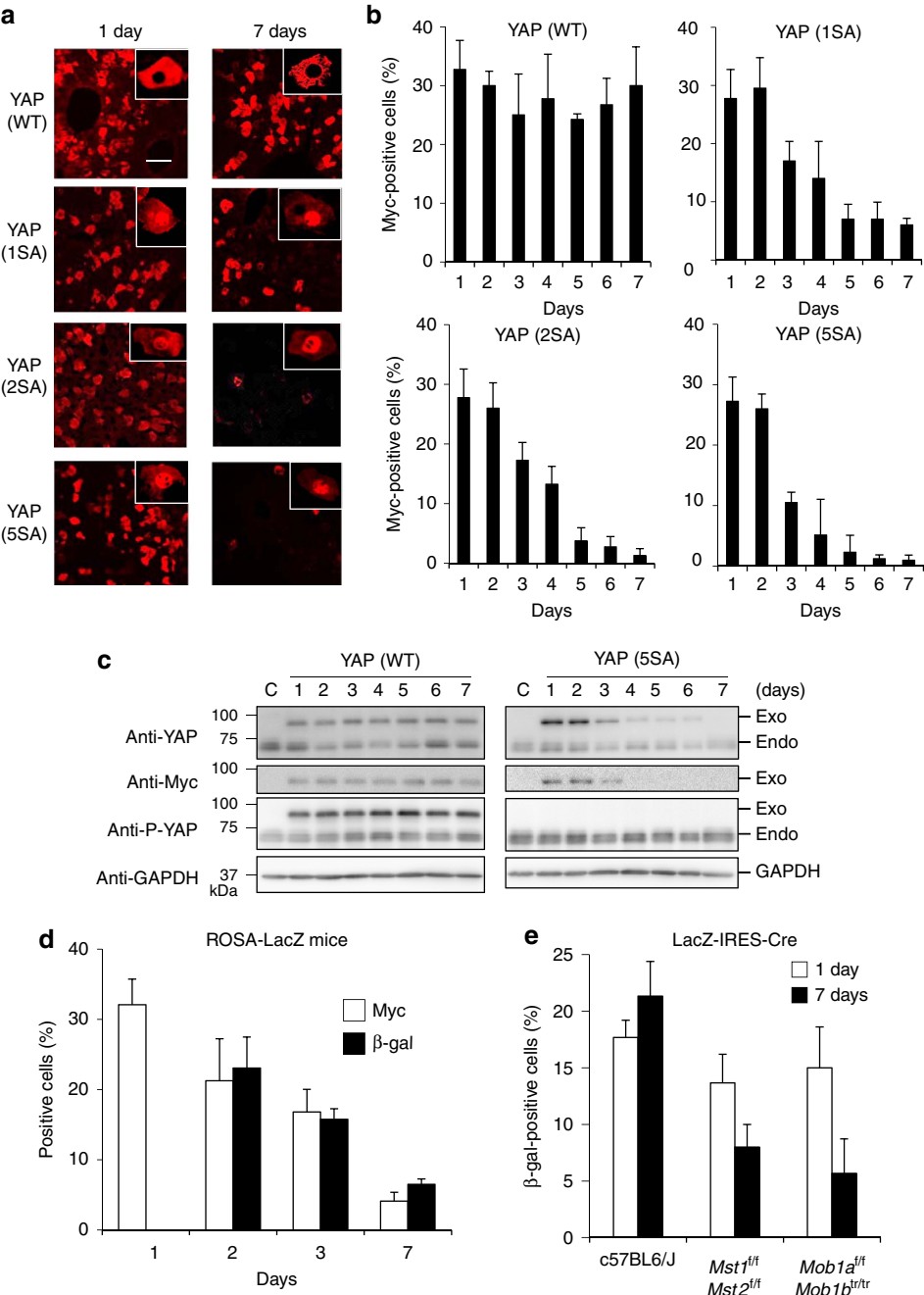

**Figure 1 | The effect of YAP activation on hepatocyte fate in mouse liver.** (**a**) Representative confocal immunofluorescence images of liver sections isolated from mice expressing Myc-tagged YAP (WT), YAP (1SA), YAP (2SA) or YAP (5SA) and stained with anti-Myc at 1 or 7 days post-HTVi (×20 objective lens). Scale bar, 100 μm ($n = 4$). Insets: High magnification images of the liver sections stained with anti-Myc (×40 objective lens). (**b**) Quantification of percentages of Myc$^+$ cells in the liver sections in (**a**) on the indicated days post-HTVi. Data are the mean ± s.d. ($n = 4$). (**c**) Immunoblot labelled with anti-YAP, anti-Myc, anti-P-YAP or anti-GAPDH to detect Myc-tagged YAP in livers of control mice (C) or mice expressing YAP (WT) or YAP (5SA) assayed on the indicated days post-HTVi. Exo, exogenous; Endo, endogenous. GAPDH, loading control ($n = 3$). Uncropped images are shown in Supplementary Fig. 12. (**d**) Quantification of Myc$^+$ and β-gal$^+$ cells in confocal immunofluorescence images of liver sections from ROSA26-LacZ mice expressing YAP (5SA)-IRES-Cre assayed on the indicated days post-HTVi. Data are the mean ± s.d. ($n = 3$) of three independent experiments. (**e**) Quantification of percentages of β-gal$^+$ cells in confocal immunofluorescence images of liver sections from Mst1$^{f/f}$;Mst2$^{f/f}$ or Mob1a$^{f/f}$;Mob1b$^{tr/tr}$ mice expressing LacZ-IRES-Cre assayed on the indicated days post-HTVi. Data are the mean ± s.d. ($n = 3$).

(5SA) expression in 10, 25, 60 or 80% of hepatocytes (Supplementary Fig. 7a,b). No matter the degree of mosaicism, and in direct contrast to the previous results using HTVi, all YAP (5SA)-expressing adenovirus-infected livers were enlarged (Fig. 3a). Staining with the proliferation marker Ki-67 revealed that this enlargement was due to increased hepatocyte

proliferation on expression of activated YAP using adenoviral infection (Fig. 3b and Supplementary Fig. 7c).

Compared to adenovirus infection, the HTVi procedure causes physical injury to liver cells. We hypothesized that this may promote the elimination of YAP-expressing cells. To test this, we injected ROSA26-LacZ mice with both Cre-

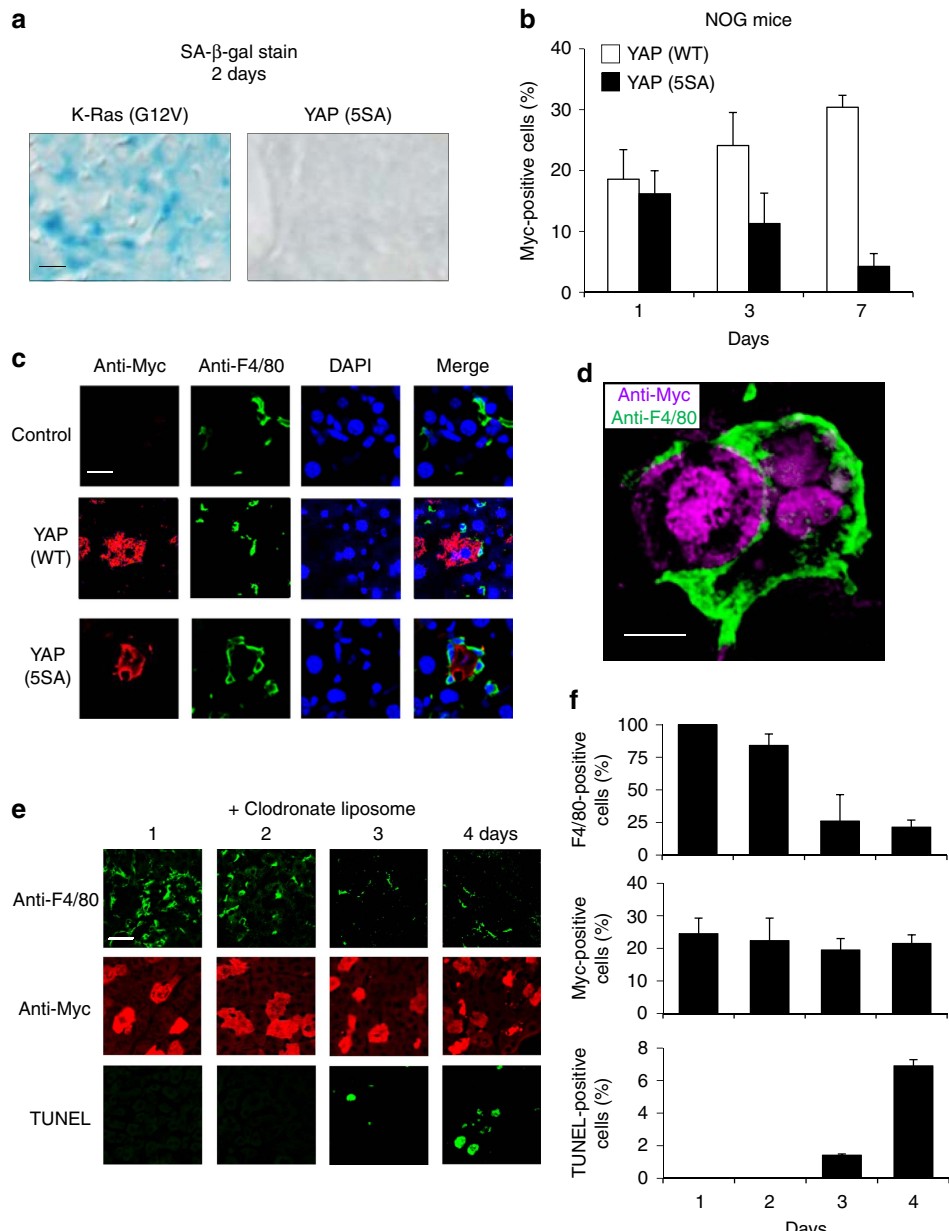

**Figure 2 | YAP activation leads to hepatocyte apoptosis and engulfment by Kupffer cells in mouse liver.** (**a**) Light microscopy to detect SA-β-gal-stained cells in liver sections of mice expressing the indicated molecules on day 2 post-HTVi. Scale bar, 10 μm ($n = 3$). (**b**) Quantification of percentages of Myc$^+$ cells in confocal immunofluorescence images of NOG mice expressing Myc-tagged YAP (WT) or YAP (5SA) assayed on the indicated days post-HTVi. Data are the mean ± s.d. ($n = 3$). (**c**) Representative confocal immunofluorescence images of the liver sections stained with anti-Myc, anti-F4/80 or 4′,6-diamidino-2-phenylindole (DAPI) on day 3 post-HTVi. Control, without HTVi. Scale bar, 20 μm ($n = 3$). (**d**) High magnification image of the liver sections stained with anti-Myc or anti-F4/80 on day 3 post-HTVi ( × 60 objective lens). Scale bar, 10 μm ($n = 3$). (**e**) Representative confocal immunofluorescence images of livers expressing Myc-tagged YAP (5SA) and treated with clodronate liposomes for the indicated days post-HTVi to deplete Kupffer cells. Sections were stained with anti-Myc, anti-F4/80 or TUNEL as indicated. Scale bar, 10 μm ($n = 4$). (**f**) Quantification of percentages of cells positive for F4/80, Myc or TUNEL in the liver sections in (**e**). Data are the mean ± s.d. ($n = 4$).

and YAP (5SA)-expressing adenovirus and subjected the animals to HTVi (Fig. 3c). HTVi indeed decreased β-gal$^+$-infected hepatocyte numbers from 50% of all cells present to 20% within 7 days. This is in contrast to YAP (5SA)-infected hepatocytes without HTVi, which increased to 70% of all cells (Fig. 3d,e and Supplementary Fig. 8). Thus, specifically in the presence of both activated YAP and cellular damage induced by HTVi, hepatocyte fate changes from proliferation to migration/apoptosis.

To investigate in more detail the type of liver injury that can change hepatocyte fate in the presence of activated YAP,

we treated Ad-Cre/Ad-YAP (5SA)-infected ROSA26-LacZ mice with either carbon tetrachloride (CCl$_4$), which causes specific injury to hepatocytes[25,26]; monocrotaline, which mainly causes LSEC injury[27]; or ethanol, which injures both LSECs and hepatocytes[28,29]. No loss of β-gal$^+$ hepatocytes was detected in livers of mice treated with CCl$_4$ or monocrotaline, but a decrease in β-gal$^+$ hepatocytes from 30 to 5% occurred in ethanol-treated livers by 7 days postinfection (Fig. 3f). Next, we administered ethanol to control mice and mice expressing YAP (WT) or YAP (5SA) via adenovirus infection. Overexpression specifically of YAP (5SA) induced substantially more hepatocyte death in the

presence of ethanol compared to YAP (WT) or control infected cells (Fig. 3g). Thus, injury to both LSECs and hepatocytes is required for significant hepatocyte elimination.

**CDC42 and Rac are required for the hepatocyte elimination.** To determine the functional domain of YAP that mediates cell elimination in injured hepatocytes, we performed a domain analysis

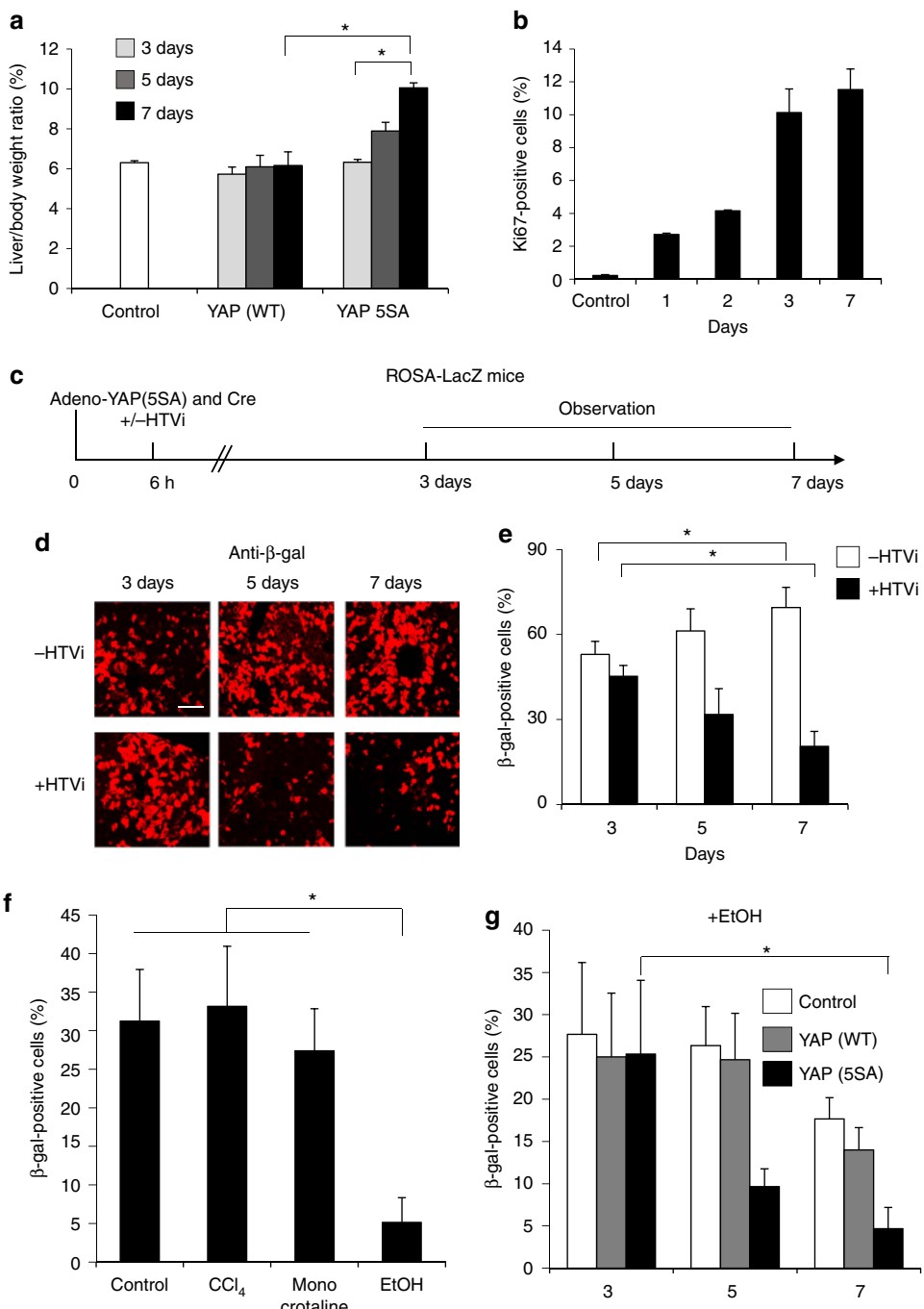

**Figure 3 | The effect of liver injury on the fate of hepatocytes expressing activated YAP.** (**a**) Quantitation of liver-to-body weight ratios of control uninfected mice or mice infected with adenovirus vector expressing YAP (WT) or YAP (5SA). Mice were killed on the indicated days postinfection. Data are the mean ± s.d. ($n = 4$). (**b**) Quantification of percentages of Ki-67$^+$ cells in liver sections from the uninfected (Control) and Adeno-YAP (5SA)-infected livers in (**a**). Data are the mean ± s.d. ($n = 4$). (**c**) Scheme showing the experimental design for analysis of the injurious effects of HTVi in ROSA26-LacZ mice infected with the indicated vectors. HTVi procedure was added at 6 h postinfection. (**d**) Representative confocal immunofluorescence images and (**e**) quantification of β-gal$^+$ cells in livers of Adeno-YAP (5SA)-infected mice, without or with HTVi, as indicated. Livers were stained with anti-β-gal at 3, 5 or 7 days postinfection. Scale bar, 10 μm. Data in (**e**) are the mean ± s.d. ($n = 4$). (**f**) Quantification of β-gal$^+$ cells in liver sections from untreated (Control), or CCl$_4^-$, monocrotaline- or ethanol (EtOH)-treated ROSA26-LacZ mice. Data are the mean ± s.d. ($n = 3$). (**g**) Quantification of β-gal$^+$ cells in liver sections from mice that were infected with YAP (WT)- or YAP (5SA)-expressing adenovirus vector and treated with EtOH. Livers were assayed on the indicated days postinfection. Data are the mean ± s.d. ($n = 3$). Statistical analyses were carried out using a paired two-tailed $t$-test and *$P < 0.05$ was considered significant.

using the following mutants: YAP (5SA/WW1,2*), defective in binding to transcription factors such as p73; YAP (1SA/ΔC) and YAP (5SA/ΔC), deletion mutants lacking the YAP activation domain; YAP (5SA/ΔPDZb), a deletion mutant lacking the PDZ-binding motif and defective in YAP nuclear localization; and YAP (5SA/TEAD*), defective in binding to TEAD[13,30,31]. YAP (5SA/WW1,2*)-expressing hepatocytes underwent the same elimination process as YAP (1SA), YAP (2SA) and YAP (5SA) hepatocytes (Fig. 4a). In contrast, hepatocytes expressing YAP (1SA/ΔC), YAP (5SA/ΔC), YAP (5SA/ΔPDZb) or YAP (5SA/TEAD*) were not lost. Thus, YAP nuclear localization, as well as TEAD binding and transcriptional activation, are indispensable for hepatocyte elimination.

To identify the molecular mechanisms involved in YAP-mediated elimination of injured hepatocytes, we analysed gene expression profiles in livers of mice expressing YAP (WT), YAP (1SA), YAP (2SA), YAP (5SA) or YAP (5SA/TEAD*) via HTVi. Using hierarchical cluster analysis, we identified gene clusters that were upregulated only in YAP (5SA) livers and not in YAP (WT) or YAP (5SA/TEAD*) livers (Supplementary Fig. 9). Examination of the gene ontology annotations of genes upregulated in YAP (1SA), YAP (2SA) and YAP (5SA) livers revealed the involvement of CDC42, which are small Rho family GTP proteins that regulate cytoskeleton organization and cell migration (Supplementary Table 1). The observed migration is not likely due to epithelial–mesenchymal transition because microarray analysis showed that epithelial–mesenchymal transition marker gene expression was not altered (Supplementary Fig. 10). We then used HTVi to introduce plasmids expressing WT CDC42 or Rac, or dominant-negative (DN) mutants of CDC42 and Rac (Fig. 4b), into control and YAP (1SA) mice[32,33]. We found that DN CDC42 and Rac, but not WT CDC42 and Rac, suppressed the elimination of YAP-activating hepatocytes by day 7 post-HTVi (Fig. 4c). Thus, both CDC42 and Rac contribute to YAP-activated hepatocyte elimination.

To identify the upstream regulators of CDC42 and Rac in hepatocytes expressing active forms of YAP, we analysed the top 60 upregulated genes from our gene ontology annotations and identified Ect2 and Fgd3, which are guanine nucleotide exchange factors for CDC42 and Rac[34–36]. Ect2 and Fgd3 mRNA levels were increased in YAP (5SA) livers but not in YAP (WT) or YAP (5SA/TEAD*) livers (Fig. 4d). Significantly, Ect2 and Fgd3 mRNAs were also induced in livers of mice infected with YAP (5SA)-expressing adenovirus and treated with ethanol or HTVi, whereas no such induction was observed in livers of mice that were infected with YAP (5SA)-expressing adenovirus and treated with CCl$_4$ (Fig. 4e). Thus, YAP activation plus HTVi or ethanol induce Ect2 and Fgd3 upregulation, which triggers CDC42 and Rac activation in hepatocytes and drives their migration to sinusoids where they undergo Kupffer cell-mediated elimination. Our data suggest a complex model for how the fate of hepatocytes in a liver subjected to myriad stresses is managed (Fig. 4f). When YAP activation was induced in hepatocytes by the adenovirus vector, the cells did not migrate or die but instead proliferated. In contrast, hepatocytes expressing activated YAP in the presence of liver injury by HTVi or ethanol migrated into sinusoids, underwent apoptosis and were engulfed by Kupffer cells. Thus, a change in hepatocyte fate from proliferation to migration/apoptosis depends on a mechanism of stress detection involving YAP.

## Discussion

The hepatocyte proliferation induced by adenovirus-mediated expression of activated YAP matches the phenotype observed in mice deficient for a Hippo pathway component and in mice transgenically expressing YAP[12–16]. In contrast, hepatocyte migration/apoptosis induced by HTVi/ethanol shown in this study is a novel cellular response. The observed apoptosis may occur through anoikis in a p73-independent manner because YAP (5SA/WW1,2*) induced hepatocyte loss. We therefore consider this hepatocyte elimination mechanism to be a primitive cellular response that is independent of adaptive immunity.

F-actin formation promotes YAP activation[37], and YAP regulates actin remodelling through the Rho GTPase-activating protein[38]. Here we found that YAP induces the Rho guanine nucleotide exchange factors Ect2 and Fgd3 in a manner dependent on an additional signal from LSECs. Thus, F-actin formation and YAP activation regulate each other through a feedback mechanism.

Cultured Madin–Darby canine kidney epithelial cells overexpressing YAP (5SA) are removed by apical extrusion when surrounded by normal Madin–Darby canine kidney cells[39]. This apical extrusion in vitro corresponds to the elimination of YAP (5SA) hepatocytes in vivo observed here. However, in vivo, hepatocyte elimination requires not only YAP activation but also liver injury.

Recently, Su et al.[40] reported interesting findings that YAP activation is insufficient to promote cellular proliferation in normal livers. They found YAP-expressing hepatocytes proliferate specifically in the presence of hepatocyte damage or inflammation. In contrast, we found a novel cell response whereby YAP-expressing hepatocytes migrate into sinusoids on both hepatocyte and LSEC injury. These data indicated that YAP-expressing hepatocytes have altered cellular dynamics depending on the status of LSECs.

In mice, impairment of the Hippo pathway or activation of YAP in intact hepatocytes leads to their proliferation and eventually hepatocellular carcinoma (Supplementary Fig. 11)[12–16]. In humans, YAP activation is observed in cases of liver fibrosis and/or liver cancer. However, our study has shown that YAP activation specifically in damaged hepatocytes triggers their elimination in normal liver. The Hippo pathway is constitutively activated and rapidly inactivates YAP by phosphorylation. Conversely, when the Hippo pathway is inactivated by stress, YAP immediately becomes unphosphorylated, translocates into the nucleus and induces target gene expression. Based on this, it is considered that YAP plays a role in an emergency stress response to maintain tissue homeostasis due to the elimination of injured cells. These findings demonstrate the complexity of cell fate determination mechanisms in vivo, and highlight a new role for YAP in tissue dynamics.

## Methods

**Mice.** C57BL/6J mice were purchased from CREA Japan and Sankyo Labo Service Corporation (Japan). ROSA26-LacZ reporter, Mst1$^{f/f}$ and Mst2$^{f/f}$ mice were purchased from the Jackson Laboratory. To generate Mob1a$^{f/f}$;Mob1b$^{tr/tr}$ mice, Mob1a$^{f/f}$ mice were mated to Mob1b$^{tr/tr}$ mice[17]. NOG mice were purchased from the Central Institute for Experimental Animals (Kanagawa, Japan). Male mice aged 8 weeks were mainly used for the experiments. All experiments were reviewed and approved by the Institutional Animal Care and Use Committee in Tokyo Medical and Dental University, and were conducted according to the committees' guidelines.

**Hydrodynamic tail vein injection.** Vectors expressing cDNAs (20 μg) were diluted in TransIT-EE Hydrodynamic Delivery Solution (Mirus Bio) to a volume equivalent to ~10% of a mouse's body weight (that is, 2 ml for a 20 g mouse). Eight-week-old male mice were placed in a restrainer, and the tail of the mice were placed in a 45–50 °C water bath for 10–20 s. Using a 2.5 ml syringe with a 27-gauge needle, the diluted vectors were injected within 7–8 s into the mouse tail veins[41–43].

**Plasmids.** The full-length human YAP cDNA was PCR-amplified and ligated to XbaI restriction sites of the expression vectors used. The YAP (1SA), YAP (2SA), YAP (5SA), YAP (1SA/ΔC), YAP (5SA/ΔC), YAP (5SA/ΔPDZb), YAP (5SA/TEAD*) and YAP (5SA/WW1, 2*) mutants were constructed by site-directed mutagenesis or deletion by PCR. The XbaI fragments of each YAP's cDNA were cloned into the pLIVE vector (Mirus Bio), in which expression of the cloned gene is driven by the hepatocyte-specific mouse AFP enhancer II and mouse minimal albumin promoter. K-Ras-active form (G12V) cDNA was cloned into pLIVE

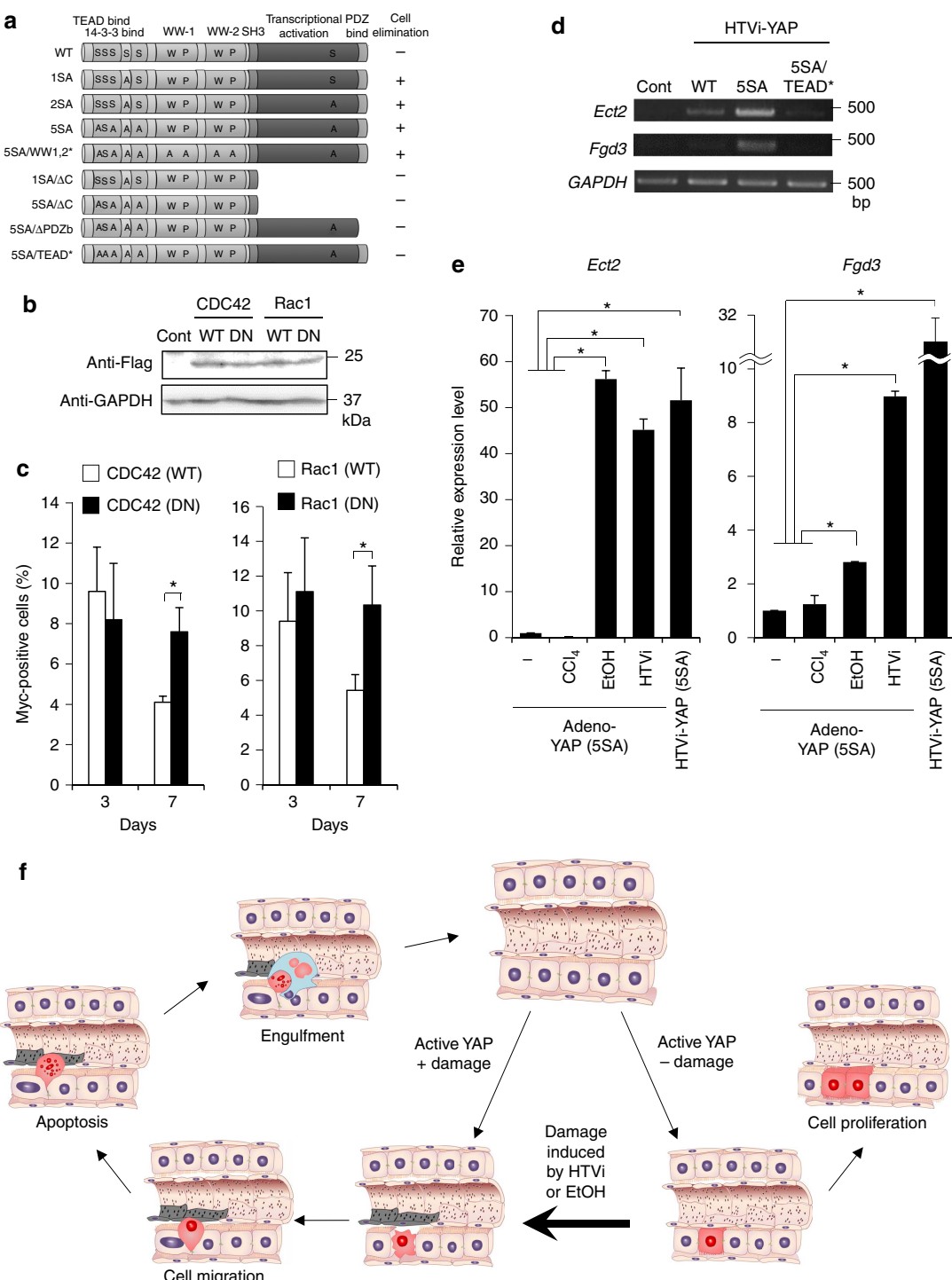

**Figure 4 | Molecular mechanism by which activated YAP induces hepatocyte migration.** (**a**) Structure and domains of human YAP WT and YAP mutants (please see main text) in plasmids that were introduced into mice by HTVi. Elimination of hepatocytes in livers was monitored as indicated ($n = 3$). (**b**) Immunoblot to detect Flag-tagged WT and DN forms of CDC42 and Rac1 in livers of mice expressing these proteins via HTVi ($n = 3$). Uncropped images are shown in Supplementary Fig. 14. (**c**) Quantification of percentages of Myc$^+$ cells in liver sections from mice expressing YAP (1SA) plus WT or DN CDC42 or Rac1. Livers were assayed on the indicated days post-HTVi. Data are the mean ± s.d. ($n = 3$). (**d**) RT–PCR analysis of Ect2 and Fgd3 mRNA levels in livers of mice expressing YAP (WT), YAP (5SA) or YAP (5SA/TEAD*) at 2 days post-HTVi. Control, no HTVi ($n = 3$). (**e**) Quantitation of mRNA levels of Ect2 and Fgd3 in livers of mice that were infected with YAP (5SA)-expressing adenovirus vector and left untreated ( − ) or treated with $CCl_4$, ethanol (EtOH) or HTVi. Real-time RT–PCR was performed on day 2 postinfection. Data are the mean ± s.d. ($n = 3$). (**f**) Schematic model of the change in mouse hepatocyte fate from proliferation to migration/apoptosis when YAP is activated in these cells by adenoviral vector infection vs inactivation of Hippo pathway by a stress such as EtOH. Statistical analyses were carried out using a paired two-tailed $t$-test and *$P < 0.05$ was considered significant.

vector. WT and dominant-negative forms of CDC42 and Rac cDNAs were separately cloned into the pCMV5 vector. IRES, β-gal and Cre were PCR amplified and ligated to form the required vectors. The absence of undesired mutations was certified by sequencing. Plasmid DNAs were purified using an EndoFree Plasmid Maxi Kit (Qiagen). The purity and quantity of plasmid DNAs were analysed by electrophoresis and absorption spectrophotometry.

**Antibodies.** The antibodies used in this study were as follows: anti-Myc (c3956; western blotting (WB):1:1,000, immunofluorescent (IF) staining:1:100) and anti-Flag (M2, F1804; WB:1:1,000) were purchased from Sigma; anti-β-gal (559761; IF:1:100) was from MP Cappel; anti-Ki-67 (Sp6: IF:1:150) was from Thermo Fisher Scientific Inc.; anti-Myc (9E10; IF:1:100) and control mouse IgG (sc-2025; IF:1:100) were from Santa Cruz; anti-LYVE1 (103-PA50; IF:1:100) was from Reliatech; anti-GAPDH (MAB374; WB:1:5,000) was from Millipore Bioscience Research Reagents; and anti-F4/80 (MCA497GA; IF:1:100) was from Serotec. Anti-P-YAP (no. 4911; WB:1:1,000) was purchased from Cell Signaling Technology. Anti-YAP (8G5; WB: 1:500; IF:1:50) and anti-Stab2 (IF:1:100) antibodies were established in our laboratory[44,45]. Actin in liver sections was stained using Alexa Fluor 488 phalloidin (A12379; IF:1:300; Thermo Fisher Scientific).

**IF staining.** Mouse livers were dissected, embedded in O.C.T. compound and sectioned at 10 μm thickness. For Ki-67 and Myc or β-gal and Myc double staining, sections were heated in 10 mmol l$^{-1}$ sodium citrate buffer (pH 6.0) at 95 °C for 10 min to facilitate antigen retrieval. Sections were preincubated with 5% bovine serum albumin and 0.1% Triton-X in PBS, followed by incubation with primary antibodies at 4 °C overnight. Sections were washed three times in PBS. Primary antibodies were visualized by incubation with secondary antibodies conjugated to Alexa Fluor 546 (mouse: A11030; rabbit: A11035; rat: A11081; 1:1,000; Thermo Fisher Scientific) or Alexa Fluor 488 (mouse: A11029; rabbit: A11034; rat: A11006; 1:1,000; Thermo Fisher Scientific) and Hoechst 33342 (H3570; 1:1,000; Thermo Fisher Scientific) for 2 h at room temperature (RT). Sections were washed three times in PBS. Sections were mounted with Mowiol 4-48 solution and images were acquired by LSM 510 Meta or LSM 710 confocal microscopy (Carl Zeiss)[39].

**Immunoblotting.** Liver extracts were fractionated by SDS–PAGE and transferred electrophoretically onto a polyvinylidene difluoride membrane. The membrane was blocked with 2 or 5% non-fat milk and incubated with each of the antibodies described in the figures for 10 h at 4 °C. Blots were incubated with the appropriate secondary antibody, peroxidase-conjugated anti-mouse, anti-rabbit or anti-rat IgG antibody (Santa Cruz), and developed with the ECL Western Blotting Detection System (Amersham Biosciences).

**X-gal and SA-β-gal stain.** Livers were sectioned at 20 μm thick, fixed with 0.5% glutaraldehyde in PBS at RT for 15 min and washed in 1 mM MgCl$_2$ in PBS. For X-gal, sections were stained overnight with 1 mg ml$^{-1}$ X-gal, 5 mM ferricyanide, 5 mM ferrocyanide and 1 mM MgCl$_2$ in PBS at 37 °C. For SA-β-gal stain, sections were stained at pH 6.0, dehydrated with ethanol and mounted. Images were acquired by Leica DMRA microscopy.

**TUNEL staining.** Liver sections were fixed with 4% paraformaldehyde and TUNEL staining was performed using the *In situ* Cell Death Detection Kit (Roche) according to the manufacturer's instructions.

**Clodronate liposome treatment.** Clodronate liposomes were prepared as below. To generate a thin layer film, 10 mg of phosphatidylcholine from egg (Avanti Polar Lipids) and 0.16 mg of cholesterol (Wako Pure Chemicals) were dissolved in 1 ml of chloroform in a glass tube, followed by evaporation of the chloroform using nitrogen gas[24]. The tube containing the film was dried in a desiccator overnight. Clodronate disodium (Sigma) dissolved by a concentration of 50 mg ml$^{-1}$ in 1 ml of PBS was added to the tube containing the film and clodronate liposomes were generated by vortexing. Control liposomes were prepared in PBS. Liposome-containing solutions were frozen and thawed three times, and subsequently passed through an extruder (Avanti Polar Lipids) with a 400 nm membrane. After centrifugation at 10 000g for 15 min, liposomes were suspended in 1 ml of sterilized PBS. To deplete macrophages, 0.1 ml clodronate liposome solution was injected into mice via the tail vein at 6 h after HTVi.

**Adenovirus infection.** The AFP enhancer, albumin promoter, YAP cDNA and poly(A) signal in the pLIVE vector were cloned into the adenovirus Cosmid vector pAxcwit2 (TaKaRa). GFP-Cre adenovirus vector was purchased from Vector BioLabs. Adenovirus preparation was performed using Vivapure AdenoPACK Kit (Sartrius) according to the manufacturer's instructions.

**Liver injury.** Eight-week-old mice were infected with adenovirus vectors. At 6 h postinfection, 1 ml kg$^{-1}$ of 30% (v v$^{-1}$) CCl$_4$ diluted in olive oil, 160 mg kg$^{-1}$ of 5% monocrotaline or 1 ml kg$^{-1}$ of 30% ethanol was administered to the mice. CCl$_4$ was administered via intraperitoneal injection, monocrotaline via intravenous injection and ethanol orally.

**Microarray analysis.** Total RNA was extracted from mouse livers using Trizol Reagent (Invitrogen) and further purified using RNeasy Mini Kits (Qiagen). The quality of RNA was initially assessed by electrophoresis on a 1.5% agarose gel, and further by absorption spectrophotometry (Agilent Bioanalyser 2100; Agilent, Palo Alto, CA, USA). cDNAs were synthesized by the Low Input Quick Amp Labelling Kit. Cy3-labelled cRNA was synthesized by *in vitro* transcription with T7 RNA polymerase. Following fragmentation, 0.6 μg cRNA was hybridized for 17 h at 65 °C on the SurePrint G3 Mouse GE 8 × 60K Microarray using the Gene Expression Hybridization Kit*4. GeneChips were washed using the Gene Expression Wash Buffers Pack and scanned using an Agilent DNA Microarray Scanner (G2565CA). Microarray data were processed using GeneChip Operating Software (Feature Extraction) and KeyMolnet Software (Institute of Medicinal Molecular Design Inc.). Gene expression microarray data have been deposited in the GEO database, with accession number GSE98231.

**Semiquantitative RT–PCR.** Total RNA was isolated from livers using TRIzol Reagent according to the manufacturer's protocol (Invitrogen). First-strand cDNA was synthesized from 1 μg total RNA using SuperscriptIII reverse transcriptase (Invitrogen) and an oligo-dT primer semiquantitative PCR was performed as below[46]. For a 20 μl PCR reaction, cDNA template was mixed with TaKaRa Ex Taq (TaKaTa) plus the appropriate primers to a final concentration of 200 nM each. The reaction was first incubated at 95 °C for 3.5 min, followed by 15–23 cycles of 95 °C for 12 s, 60 °C for 13 s and 72 °C for 30 s. PCR primers for the Ect2, Fgd3 and GAPDH genes were as follows: Ect2 FW, 5′-GATTAAGAAGGTGCTGGAC-ATCCG-3′ and Ect2 RW, 5′-AGAAAGAAGGGAGGGCTGACAAGGG-3′; Fgd3 FW, 5′-AAGCTGCTTCATATTGCCCAGGAG-3′ and Fgd3 RW, 5′-AGTCCTTG-AGCAGCAGCTCATAGC-3′; GAPDH FW, 5′-GCATCCACTGGTGCTGCC-AAGGC-3′ and GAPDH RW, 5′-TAGGCCCCTCCTGTTATTATGG-3′.

**Quantitative RT–PCR.** Quantitative real-time PCR with reverse transcription (RT–PCR) was performed as below[47]. For a 20 μl PCR reaction, cDNA template was mixed with 10 μl SsoFast EvaGreen Supermix (Bio-Rad) plus the appropriate primers to a final concentration of 200 nM each. The reaction was first incubated at 95 °C for 3.5 min, followed by 41 cycles of 95 °C for 12 s, 60 °C for 13 s and 72 °C for 18 s. PCR primers for the Ctgf, Ect2, Fgd3 and GAPDH genes were as follows: Ctgf FW, 5′-GTGTGCACTGCCAAAGATGGTGC-3′ and Ctgf RW, 5′-GCAC-GTCCATGCTGCACAG-3′ and Ect2 FW, 5′-GATTAAGAAGGTGCTGGA-CATCCG-3′ and Ect2 RW, 5′-AGCATCTGCCTTACAAATGGTGTTGGC-3′; Fgd3 FW, 5′-AAGCTGCTTCATATTGCCCAGGAG-3′ and Fgd3 RW, 5′-TCGGGCAGGAGGAACTGCCCATGG-3′; GAPDH FW, 5′-AACTCGGCC-CCCAACACT-3′ and GAPDH RW, 5′-TAGGCCCCTCCTGTTATTATGG-3′.

**Statistical analyses.** Sample sizes were determined on the basis of pilot experiments and previous experience from similar experiments. To examine whether the data had the same variances, we analysed them by F-test. As all the data were determined to be normally distributed, parametric statistics were used throughout. Data were analysed by Student's t-test or Welch's t-test. All the t-tests were performed as two-tailed t-tests. The statistical test used for each experiment is stated in the figure legend.

**Data availability.** The authors declare that all data supporting the findings of this study are available within the paper and its Supplementary Information files or are available from the corresponding author on request.

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

## Acknowledgements

We thank numerous members of the Nishina laboratory and Helen Pickersgill of Life Science Editors for their helpful discussions and critical comments on the manuscript. This work is supported by a Japan Society for the Promotion of Science (JSPS) Grant-in-Aid for Scientific Research on Innovative Areas 26114001, a Grant-in-Aid from the Ministry of Health, Labour and Welfare of Japan and a Grant-in-Aid from the Uehara Memorial Foundation.

## Author contributions

N.M. and H.N. designed experiments and wrote the manuscript. N.M. and S.H. performed experiments. T.I., M.T. and A.M. contributed to the design of the mouse gene expression experiments. M.I. and Y.O. contributed to the design and execution of the clodronate liposome experiments. M.N. and A.S. contributed to the design and execution of the knockout mouse experiments. S.T. and I.S. contributed to the design of the liver injury experiments.

## Additional information

**Competing interests:** The authors declare no competing financial interests.

DOI: 10.1038/ncomms16146    OPEN

# Erratum: YAP determines the cell fate of injured mouse hepatocytes *in vivo*

Norio Miyamura, Shoji Hata, Tohru Itoh, Minoru Tanaka, Miki Nishio, Michiko Itoh, Yoshihiro Ogawa, Shuji Terai, Isao Sakaida, Akira Suzuki, Atsushi Miyajima & Hiroshi Nishina

*Nature Communications* 8:16017 doi: 10.1038/ncomms16017 (2017); Published 6 Jul 2017; Updated 7 Aug 2017.

The financial support for this Article was not fully acknowledged. The Acknowledgements should have included the following:

This work was supported by a Nanken-Kyoten grant from Tokyo Medical and Dental University (TMDU).

