## [Peer review file · Nature Communications]

Reviewers' comments:

Reviewer #1 (Remarks to the Author):

In this study, the authors investigated the consequences of overexpressing YAP active mutant constructs in hepatocytes of adult mice. They found that YAP activation by adenoviral infection leads to hepatocytes proliferation, which is in line with previously published observations. By contrast, when they used hydrodynamic tail injection of plasmid DNAs, the YAP-overexpressing cells underwent selective elimination, which seems to be mediated by the Kupffer cells in hepatic sinusoids. Similar observations were made in other types of liver damage, such as exposure to ethanol. Based on these findings, the authors concluded that YAP might act as a "stress sensor" promoting elimination of injured cells to maintain tissue and organ homeostasis. They also made an attempt to dissect the mechanisms downstream of YAP activation, suggesting that this involves the GEFs, Ect2 and Fgd3, which in turn activate Cdc42 and Rac to regulate cell migration of YAP-activated hepatocytes into the sinusoids.

The idea that hepatocytes can switch between a proliferative state to migration/apoptosis based on activated YAP and injury is fascinating. Nevertheless, this reviewer is not sure in which physiological or pathological context(s) this mechanism might be relevant. Besides, there are many flaws with this study and the data are of mixed quality in many places that do not justify the conclusions made. Specific concerns with the manuscript are listed below.

1) This study is mostly based on hydrodynamic tail vein injection to overexpress plasmid DNAs in the liver. One of the major limitations of this approach is that the overexpressed genes are rapidly degraded in hepatocytes. Usually, the expression levels of the transgene peak approximately 8 to 24 hours after injection, and undergo a dramatic decrease over 7 days (Zhang et al. Gene Ther. 2004). To achieve a more stable gene expression, one should use transposon plasmids. Thus, the authors should provide here a thorough characterization of the system used and overexpression of YAP WT and mutant constructs in the liver. For instance, Western Blot showing later time points (after day 4) and hybridized against Myc should be included? Is there a true activation of the YAP signaling in the injected-hepatocytes? What are the YAP-phosphorylation status and downstream Lats and TEAD activity?

2) The authors used Mob and Mst floxed mouse strains as an additional system to inactivate the Hippo pathway upon injection of LacZ-IRES-Cre (see Fig. 1e). But, again there is no direct functional demonstration that the effects of Mob or Mst depletions are due to YAP activation. The status of the Hippo pathway components should also be probed in the liver following Mob/Mst depletion and YAP-overexpression (see point#1).

3) The correlation between proliferation and loss of YAP-(5SA)-overexpressing hepatocyte is very indirect. At least, data showing beta-gal staining coupled with anti-Myc staining in the liver should be shown. Also, the % of Myc- and Ki67-double-positive cells should be counted and included in the manuscript. In addition, a beta-gal staining needs to be performed to corroborate the immunostaining with anti-beta-Gal antibody, which rarely works.

4) Liver injury is a very important point for the conclusion of this paper. The authors claim that that hydrodynamic injection leads to liver injury. This is true, even though the injury is transient and the liver heals in approximately 1 week (Zhang et al. Gene Ther. 2004). There are also some discrepancies in the results they obtained with the different types of liver injury. Why only hydrodynamic injection and alcohol administration induce "YAP-induced hepatocyte fate to switch between proliferation to migration/apoptosis"? In addition, the lack of phenotype upon the LSEC insult with Monocrotaline (Fig. 3) is not consistent with the results shown after the depletion of Kupffer cells (Fig. 4).

5) The characterization of the mechanisms possibly underlying the migration phenotype is very preliminary and numerous controls are missing. First, cellular migration is never shown in the

study but only inferred. Second, the status of Cdc42 and Rac activation in the liver upon overexpression of the WT and DN plasmid is not assessed. Overall, the analysis of the mechanism needs to be complemented with an in vitro system allowing to use for example fret sensor and to study migration. This could be done in mouse primary hepatocytes or well-established hepatocyte cell lines.

6) The figure legends of the Supplementary Figures are missing. Many Supplementary Figures are not cited in the main text.

Reviewer #2 (Remarks to the Author):

In this manuscript, the authors perform an in vivo mosaic analysis of YAP overexpressing hepatocytes in the mouse liver. They show that liver injury leads to the elimination of hepatocytes with hyperactive YAP. Interestingly, the elimination of YAP-activated hepatocytes only takes place when liver sinusoidal endothelial cells are injured, such as after hydrodynamic injection or EtOH treatment. In contrast, YAP activation in undamaged hepatocytes or liver injury that doesn't affect sinusoids doesn't lead to hepatocyte elimination and results in hepatocyte proliferation. A combination of structure function analysis of YAP and microarray profiling of gene expression identified that the transcriptional activity of YAP is required for the elimination of hepatocytes, and that expression of *Ect2* and *Fgd3*, which encode for proteins that activate CDC42 and Rac, is required to promote cell migration, sinusoid intra-vasation and elimination of hepatocytes by apoptosis and engulfment by Kupffer cells.

Thus, the paper describes a novel mechanism where the levels of YAP activity determine the fate and behavior of hepatocytes in an injury type dependent manner. These findings are relevant for regenerative medicine and show that not all YAP expressing cells have the potential to regenerate injured organs.

Although most conclusions are well supported by data, there are still some issues that need to be further addressed for publication.

- Su et al. (eLife, 2015), used a mosaic mouse model to show that Yap activation is insufficient to promote growth in normal livers. In their model, the fraction of hepatocytes expressing Yap in non injured livers do not expand clonally and present increased apoptosis. In response to injury or inflammation, however, YAP expressing hepatocytes expand clonally. How can the authors reconcile the findings of the paper with the ones of Su et al.? This should be addressed in the discussion of the paper.

- In Figure 1A, the quality of the pictures and the magnification are too low to distinguish the localization of myc-tagged YAP. Strikingly, the localization of all forms of YAP (normal and hyperactive) is the same, and there is no nuclear enrichment in the hyperactive mutants. One would expect to see nuclear localization of YAP 1SA, 2SA and 5SA. A higher quality magnification picture should be provided where the nuclei are co-stained. Same problem can be seen in other figures. The same comment applies for all other figures.

- In Figure 2E, it is not clear which type of cells are the apoptotic cells. A merged picture will be helpful to distinguish if the apoptotic cells are the YAP-expressing cells or not.

- In Figure 3, the authors use Adeno-YAP(5SA) + cre on Rosa LacZ mice as a surrogate reporter. However, the b-Gal staining does not reflect YAP levels or activity, so in the absence of a double staining of YAP and b-Gal one cannot conclude that the b-Gal positive cells have higher levels of YAP. The same for the KI67 + cells, are they the b-Gal expressing cells? Double staining is required. This is a very important issue and needs to be documented in the paper.

- In Sup.Figure 2, the livers should be stained with a LSEC marker, such as LIVE1 to determine if hepatocytes are in the sinusoid or not.

Reviewer #3 (Remarks to the Author):

In this manuscript Nishina et al. use in vivo mosaic analysis to show that YAP activation in injured hepatocytes induces their elimination, whereas leads to proliferation in undamaged hepatocytes.

They further show that the process is independent of adaptive immunity and senescence surveillance. They instead describe the gradual process of Yap/Hippo dependent elimination of damaged hepatocytes as a sequence of events that involves migration of injured YAP-activated hepatocytes to the hepatic sinusoids, apoptosis, and engulfment by Kupffer cells.

Cellular stresses that induce damage to hepatocytes change cell fate from proliferation to migration/apoptosis, through a process that involves GEF, Ect2, Fgd3, CDC42 and Rac to regulate cell migration, as well as activated YAP. The results suggests that YAP acts different in healthy and injured hepatocytes, probably due to preexisting changes in cytoskeletal regulators in damaged cells. The results are interesting and novel, but the MS is written is conceptually confusing and should be rewritten.

Major comments:

- I had more trouble than normal reading this paper because they use concepts that are different (like senescent cells versus unfit cells) in what I found to be a rather confusing way. For example, the first sentence "Cellular stress in tissues and organs leads to unfit cells that are damaged, senescent or transformed" is confusing and I am not sure it is conceptually correct. Unfit cells is now a biological term normally used to describe cells that are viable on their own, and able to even create a normal individual, but recognized and eliminated by apoptosis in the context of wildtype or fitter cells. Senescent cells, on the contrary, are not able to generate normal individuals and is an alternative fate to apoptosis.

- Pathways involved in senescence are not linked to what they study (as they show in their MS using senescence markers), so the use of the term senescent cells in the intro is not bringing any clarity. Also, this confusion, means that the citations they quote many times do not refer to what they mention in the text: references 1 to 4 do not deal with the concept of unfit cells but rather with the concept of senescent cells and the pathways inducing senescence. What they see is apoptosis and engulfment of unfit cells, nothing to do with the terminal fate of senescence. I think many of those conceptual problems need to be solved before it is a readable piece and the reader can make sense of their data. The many conceptual confusions the authors have will only confuse the reader and the field.

- The data that suggest that YAP acts different in healthy and injured hepatocytes, and could be a mechanism to eliminate damaged hepatocytes are interesting. It is however unclear whether this is a cell autonomous process or requires interactions with neighboring cells and hence comparison of cell fitness. It is possible that preexisting changes in cytoskeletal regulators in damaged cells modulate YAP function in a cell autonomous manner, without fitness comparison between cells. I think it is therefore also confusing to use the term unfit cells. It will be easier to just mention damaged cells.

- At this stage it is also unclear what are the physiological consequences of this mechanism. Although I understand this is not easy to asses, it would be good to mention this uncertainty in the discussion. How important is this mechanism to maintain organ function?

II. Specific Responses to Reviewer #1:

In this study, the authors investigated the consequences of overexpressing YAP active mutant constructs in hepatocytes of adult mice. They found that YAP activation by adenoviral infection leads to hepatocytes proliferation, which is in line with previously published observations. By contrast, when they used hydrodynamic tail injection of plasmid DNAs, the YAP-overexpressing cells underwent selective elimination, which seems to be mediated by the Kupffer cells in hepatic sinusoids. Similar observations were made in other types of liver damage, such as exposure to ethanol. Based on these findings, the authors concluded that YAP might act as a “stress sensor” promoting elimination of injured cells to maintain tissue and organ homeostasis. They also made an attempt to dissect the mechanisms downstream of YAP activation, suggesting that this involves the GEFs, Ect2 and Fgd3, which in turn activate Cdc42 and Rac to regulate cell migration of YAP-activated hepatocytes into the sinusoids.

The idea that hepatocytes can switch between a proliferative state to migration/apoptosis based on activated YAP and injury is fascinating. Nevertheless, this reviewer is not sure in which physiological or pathological context(s) this mechanism might be relevant. Besides, there are many flaws with this study and the data are of mixed quality in many places that do not justify the conclusions made. Specific concerns with the manuscript are listed below.

We thank this reviewer for his/her positive comments.

We respond to all the raised issues as described below.

1) This study is mostly based on hydrodynamic tail vein injection to overexpress plasmid DNAs in the liver. One of the major limitations of this approach is that the overexpressed genes are rapidly degraded in hepatocytes. Usually, the expression levels of the transgene peak approximately 8 to 24 hours after injection, and undergo a dramatic decrease over 7 days (Zhang et al. Gene Ther. 2004). To achieve a more stable gene expression, one should use transposon plasmids. Thus, the authors should provide here a thorough characterization of the system used and overexpression of YAP WT and mutant constructs in the liver. For instance, Western Blot showing later time points (after day 4) and hybridized against Myc should be included? Is there a true activation of the YAP signaling in the injected-hepatocytes? What are the YAP-phosphorylation status and downstream Lats and TEAD activity?

We thank the reviewer for his/her insightful comments.

Accordingly, we performed Western blots during 1-7 days post-HTVi to detect exogenous and endogenous YAP by using anti-YAP, anti-Myc and anti-phospho

(P)-YAP antibodies. We found that exogenous YAP (WT) expression remained stable for the full 7 days (Revised Fig. 1c). The phosphorylated form of exogenous YAP (WT) was also stable. Thus, even with our gene expression system, we were fortunately able to achieve stable expression of exogenous YAP. In contrast, exogenous YAP mutant (1SA, 2SA and 5SA) expression decreased during days 3-7 post-HTVi, in parallel with the observed loss of mutant YAP-expressing hepatocytes (Revised Fig. 1c and Supplementary Fig. 1c).

Upstream, the Hippo mammalian homologue, Mst, phosphorylates Lats, which in turn phosphorylates YAP. Phosphorylated YAP remains in the cytoplasm in an inactive form, whereas unphosphorylated, active YAP translocates into the nucleus, interacts with TEAD and induces target gene expression such as of *connective tissue growth factor* (*ctgf*). We verified the activation of YAP signaling by analyzing YAP nuclear localization and *ctgf* gene expression. We found that the active YAP mutants (1SA, 2SA and 5SA) but not YAP (WT) localized in the nucleus and induced *ctgf* gene expression (Revised Fig. 1a insets and Supplementary Fig. 1b). Thus, YAP signaling is indeed activated in the injected-hepatocytes.

2) The authors used Mob and Mst floxed mouse strains as an additional system to inactivate the Hippo pathway upon injection of LacZ-IRES-Cre (see Fig. 1e). But, again there is no direct functional demonstration that the effects of Mob or Mst depletions are due to YAP activation. The status of the Hippo pathway components should also be probed in the liver following Mob/Mst depletion and YAP-overexpression (see point#1).

We thank the reviewer for this comment.

We have recently shown that Mob depletion in mouse liver decreased YAP-phosphorylation and enhanced YAP mRNA and protein levels, YAP nuclear translocation, *ctgf* gene expression and liver cancer formation (revised reference #17). These Mob-deficient liver phenotypes were dramatically suppressed by additional YAP loss. Thus, the liver phenotypes of Mob depletion are strongly dependent on YAP. We now describe these results more clearly in the Introduction.

Furthermore, we measured *ctgf* expression levels over time in mouse livers treated with LacZ-IRES-Cre by HTVi (to deplete Mob/Mst and thereby activate the YAP pathway). *ctgf* gene expression was up-regulated in livers lacking Mob1a/Mob1b or Mst1/Mst2 at 2 and 3 days post-HTVi, further verifying that YAP is indeed activated in this system (Revised Supplementary Fig. 4.)

3) The correlation between proliferation and loss of YAP-(5SA)-overexpressing

hepatocyte is very indirect. At least, data showing beta-gal staining coupled with anti-Myc staining in the liver should be shown. Also, the % of Myc- and Ki67-double-positive cells should be counted and included in the manuscript. In addition, a beta-gal staining needs to be performed to corroborate the immunostaining with anti-beta-Gal antibody, which rarely works.

To address these concerns, we performed additional β -Gal staining with X-Gal, and co-immunostaining with anti- β -Gal and anti-Myc antibodies as requested.

To show β -Gal expression coupled with Myc-YAP (5SA) expression in hepatocytes, we performed the X-Gal stain or anti- β -Gal and anti-Myc double stain in YAP (5SA)-IRES-Cre expressing ROSA mice. We found that, consistent with the anti- β -Gal data, X-Gal-stained hepatocytes decreased after 3 days post-HTVi (Revised Supplementary Fig. 2a). Importantly, β -Gal-positive hepatocytes coincided with Myc-positive hepatocytes, which also decreased after 3 days post-HTVi (Revised Supplementary Fig. 2b). These results indicate that YAP (5SA)-expressing hepatocytes could be labeled by β -Gal.

To investigate the correlation between proliferation and loss of YAP (5SA)-expressing hepatocytes, we stained with anti-Myc and anti-Ki67 antibodies and quantified the percentage of Ki67-positive cells in YAP (5SA)-expressing liver on 7 days post-HTVi. We found that the Ki67-positive cells were predominantly (~90%) Myc-negative hepatocytes (Revised Supplementary Fig. 3d). These results indicated that the YAP (5SA)-expressing cells were lost and YAP (5SA)-negative hepatocytes proliferated.

4) Liver injury is a very important point for the conclusion of this paper. The authors claim that that hydrodynamic injection leads to liver injury. This is true, even though the injury is transient and the liver heals in approximately 1 week (Zhang et al. Gene Ther. 2004). There are also some discrepancies in the results they obtained with the different types of liver injury. Why only hydrodynamic injection and alcohol administration induce “YAP-induced hepatocyte fate to switch between proliferation to migration/apoptosis”? In addition, the lack of phenotype upon the LSEC insult with Monocrotaline (Fig. 3) is not consistent with the results shown after the depletion of Kupffer cells (Fig. 4).

We apologise for not explaining this clearly enough.

YAP-induced hepatocyte elimination is induced upon injury to both hepatocytes and liver sinusoidal endothelial cells (LSECs) at the initial stage (revised Fig. 3 and Fig. 4f). HTVi and EtOH injure both LSECs and hepatocytes, however, CCl₄ injures only hepatocytes and monocrotaline injures only LSECs. Therefore, only hydrodynamic

injection and alcohol administration induce the hepatocyte elimination.

The clodronate treatment (i.e., Kupffer cell depletion) was used to show the engulfment of damaged YAP (5SA)-expressing hepatocytes by Kupffer cells at the final stage (Revised Figs. 2 and Fig. 4f). Thus, the LSEC insult and the engulfment by Kupffer cells occur at different stages, respectively, so they would not be expected to be consistent. We have tried to make this clearer in the revised manuscript.

5) The characterization of the mechanisms possibly underlying the migration phenotype is very preliminary and numerous controls are missing. First, cellular migration is never shown in the study but only inferred. Second, the status of Cdc42 and Rac activation in the liver upon overexpression of the WT and DN plasmid is not assessed. Overall, the analysis of the mechanism needs to be complemented with an *in vitro* system allowing to use for example fret sensor and to study migration. This could be done in mouse primary hepatocytes or well-established hepatocyte cell lines.

We thank the reviewer for his/her important comment.

We used LSEC staining to show directly that YAP (5SA) hepatocytes migrate from monolayer hepatic cords to hepatic sinusoids, “sinusoid intra-vasation of hepatocytes” (revised Supplementary Fig. 5a-c). This was inhibited by dominant negative (DN) forms of CDC42 and Rac (revised Fig. 4c). To show hepatocyte migration into LSEC more clearly, we have also used another antibody, anti-LYVE1, which is also specific for LSEC (Revised Supplementary Fig. 5d).

We understand the importance of assessing CDC42 and Rac activation directly in the liver. However, it was technically impossible to apply FRET or antibodies recognizing these active forms in mouse living liver. Therefore, we used DN forms of Rac1 and CDC42, which have been used successfully *in vivo* (revised references #31 and 32). Consistent with our model, we found that DN forms of CDC42 and Rac suppressed YAP-inducing hepatocytes elimination (revised Fig. 4c). These further demonstrate that the DN forms represent activated CDC42 and Rac.

We also attempted *in vitro* experiments using mouse primary hepatocytes and the hepatocyte cell line, HepG2. However, these cells do not form the relevant hepatic cord structures and are thus unable to recapitulate hepatocyte elimination/migration.

6) The figure legends of the Supplementary Figures are missing. Many Supplementary Figures are not cited in the main text.

Perhaps there was a problem with the database here - all figure legends of the original Supplementary Figures (Extended Data Figures 1-6) were described on pages 22 and 23,

and all original Supplementary Figures were also cited in the original MS.

We have again ensured that we describe all the figure legends of the revised Supplementary Figures, and cited all revised Supplementary Figures in the revised MS.

III. Specific Responses to Reviewer #2:

In this manuscript, the authors perform an in vivo mosaic analysis of YAP overexpressing hepatocytes in the mouse liver. They show that liver injury leads to the elimination of hepatocytes with hyperactive YAP. Interestingly, the elimination of YAP-activated hepatocytes only takes place when liver sinusoidal endothelial cells are injured, such as after hydrodynamic injection or EtOH treatment. In contrast, YAP activation in undamaged hepatocytes or liver injury that doesn't affect sinusoids doesn't lead to hepatocyte elimination and results in hepatocyte proliferation. A combination of structure function analysis of YAP and microarray profiling of gene expression identified that the transcriptional activity of YAP is required for the elimination of hepatocytes, and that expression of Ect2 and Fgd3, which encode for proteins that activate CDC42 and Rac, is required to promote cell migration, sinusoid intra-vasation and elimination of hepatocytes by apoptosis and engulfment by Kupffer cells.

Thus, the paper describes a novel mechanism where the levels of YAP activity determine the fate and behavior of hepatocytes in an injury type dependent manner. These findings are relevant for regenerative medicine and show that not all YAP expressing cells have the potential to regenerate injured organs.

We thank this reviewer for his/her positive comments.

We respond to all the raised issues as described below.

Although most conclusions are well supported by data, there are still some issues that need to be further addressed for publication.

•Su et al. (eLife, 2015), used a mosaic mouse model to show that Yap activation is insufficient to promote growth in normal livers. In their model, the fraction of hepatocytes expressing Yap in non injured livers do not expand clonally and present increased apoptosis. In response to injury or inflammation, however, YAP expressing hepatocytes expand clonally. How can the authors reconcile the findings of the paper with the ones of Su et al.? This should be addressed in the discussion of the paper.

We thank the reviewer for this interesting comment.

Su et al. reported that YAP-expressing hepatocytes did not proliferate in normal liver. Interestingly, they found that YAP-expressing hepatocytes do proliferate in the presence of hepatocyte damage by CCl₄ or inflammation by LPS. In contrast, we found a novel cell response that YAP-expressing hepatocytes migrate into sinusoids (sinusoid intra-vasation of hepatocytes) and undergo apoptosis in the presence of hepatocyte and LSEC damages by HTVi or EtOH. Thus, the difference between the two experimental

conditions is with/without LSEC injury.

The proliferation of YAP-activating hepatocyte was induced by adenovirus infection. It has been reported that inflammatory cytokines such as IL-6 and TNF are induced by adenovirus infection (Q Liu and DA Muruve. *Gene Therapy* 2003). So, it is possible that YAP (5SA)-expressing hepatocytes proliferate upon inflammation.

We have now added sentences in the “Discussion” and cited the paper (revised reference #39).

“Recently, Su et al. reported interesting findings that YAP activation is insufficient to promote cellular proliferation in normal livers³⁹. They found that YAP-expressing hepatocytes proliferate specifically in the presence of hepatocyte damage or inflammation. In contrast, we found a novel cell response whereby YAP-expressing hepatocytes migrate into sinusoids upon both hepatocyte and LSEC injury. These data indicated that YAP-expressing hepatocytes have altered cellular dynamics depending on the status of LSECs.”

-In Figure 1A, the quality of the pictures and the magnification are too low to distinguish the localization of myc-tagged YAP. Strikingly, the localization of all forms of YAP (normal and hyperactive) is the same, and there is no nuclear enrichment in the hyperactive mutants. One would expect to see nuclear localization of YAP 1SA, 2SA and 5SA. A higher quality magnification picture should be provided where the nuclei are co-stained. Same problem can be seen in other figures. The same comment applies for all other figures.

We thank the reviewer for his/her insightful comment.

To observe the cell responses over a wide region in mouse liver, we used high quality and low magnification images taken with the 20x objective lens. As a result, we could observe the loss of active YAP-expressing hepatocytes. On the other hand, as the reviewer pointed out, information on the localization of all forms of YAP was lost.

Therefore, we have now added new data with higher magnification images taken using the 40x objective lens (revised Fig. 1a insets). YAP (WT) localized in the cytoplasm at days 1 and 7. As expected, active YAP (1SA, 2SA and 5SA) localized in the nucleus at days 1 and 7.

The original Fig. 2d demonstrated engulfment of YAP (5SA)-expressing hepatocytes by Kupffer cells. The DAPI stain (blue) blocked the nuclear localization of YAP (5SA). So, we have deleted the DAPI stain in revised Fig. 2d, which again clearly shows the nuclear localization of YAP (5SA). Furthermore, we have added a new 3D movie showing with/without DAPI stain (revised Supplementary Movie 1).

-In Figure 2E, it is not clear which type of cells are the apoptotic cells. A merged picture will be helpful to distinguish if the apoptotic cells are the YAP-expressing cells or not.

To show that the apoptotic cells are indeed the YAP-expressing cells, we performed the TUNEL staining coupled with anti-Myc staining in the clodronate treated YAP (5SA) expressing livers at 1-4 days post-HTVi. The results show that all TUNEL positive cells are Myc-positive cells, 3 and 4 days post-HTVi (revised Supplementary Fig. 6). Thus, apoptosis occurs in the YAP (5SA)-expressing hepatocytes.

-In Figure 3, the authors use Adeno-YAP(5SA) + cre on Rosa LacZ mice as a surrogate reporter. However, the b-Gal staining does not reflect YAP levels or activity, so in the absence of a double staining of YAP and b-Gal one cannot conclude that the b-Gal positive cells have higher levels of YAP. The same for the KI67 + cells, are they the b-Gal expressing cells? Double staining is required. This is a very important issue and needs to be documented in the paper.

We thank the reviewer for his/her insightful comment.

To show clear images of the hepatocytes, we used red and green double fluorescence stains. We used GFP-fused Cre adenovirus vector, so β -Gal, Myc-YAP and Ki67 were stained red.

The reviewer requests double stain of anti- β -Gal and anti-Myc (Myc-YAP) or anti- β -Gal and anti-Ki67 (cellular proliferation). So, we first performed double stain of GFP (green) and anti- β -Gal (red). We found that β -Gal positive hepatocytes were identical to GFP positive cells 3-7 days post-infection (revised Supplementary Fig. 8a). Next, we performed a double stain of GFP (green) and anti-Myc (red), or GFP (green) and anti-Ki67 (red). Almost all GFP-positive hepatocytes were also Myc-positive cells (revised Supplementary Fig. 8b). We also found overlap between Ki67-positive hepatocytes and GFP-positive cells (Revised Supplementary Fig. 8c). These results show that β -Gal staining reflects YAP expression, and the YAP-expressing hepatocytes proliferated.

-In Sup.Figure 2, the livers should be stained with a LSEC marker, such as LIVE1 to determine if hepatocytes are in the sinusoid or not.

Accordingly, we performed LYVE1 stain.

We stained YAP (WT) and (5SA)-expressing mouse livers at 3 day post-HTVi by anti-LYVE1, anti-YAP and DAPI. We found that YAP (WT)-expressing cells located to hepatic cords (revised Supplementary Fig. 5d). In contrast, YAP (5SA)-expressing

hepatocytes were surrounded by LYVE1-positive cells. These results showed that YAP (5SA)-expressing hepatocytes migrate to the sinusoid.

IV. Specific Responses to Reviewer #3:

In this manuscript Nishina et al. use in vivo mosaic analysis to show that YAP activation in injured hepatocytes induces their elimination, whereas leads to proliferation in undamaged hepatocytes.

They further show that the process is independent of adaptive immunity and senescence surveillance. They instead describe the gradual process of Yap ∇ Hippo dependent elimination of damaged hepatocytes as a sequence of events that involves migration of injured YAP-activated hepatocytes to the hepatic sinusoids, apoptosis, and engulfment by Kupffer cells.

Cellular stresses that induce damage to hepatocytes change cell fate from proliferation to migration/apoptosis, through a process that involves GEF, Ect2, Fgd3, CDC42 and Rac to regulate cell migration, as well as activated YAP. The results suggests that YAP acts different in healthy and injured hepatocytes, probably due to preexisting changes in cytoskeletal regulators in damaged cells. The results are interesting and novel, but the MS is written is conceptually confusing and should be rewritten.

We thank this reviewer for his/her positive comments.

We have rewritten the revised MS as described below.

Major comments:

- I had more trouble than normal reading this paper because they use concepts that are different (like senescent cells versus unfit cells) in what I found to be a rather confusing way. For example, the first sentence "Cellular stress in tissues and organs leads to unfit cells that are damaged, senescent or transformed" is confusing and I am not sure it is conceptually correct. Unfit cells is now a biological term normally used to describe cells that are viable on their own, and able to even create a normal individual, but recognized and eliminated by apoptosis in the context of wildtype or fitter cells. Senescent cells, on the contrary, are not able to generate normal individuals and is an alternative fate to apoptosis.

- Pathways involved in senescence are not linked to what they study (as they show in their MS using senescence markers), so the use of the term senescent cells in the intro is not bringing any clarity. Also, this confusion, means that the citations they quote many times do not refer to what they mention in the text: references 1 to 4 do not deal with the concept of unfit cells but rather with the concept of senescent cells and the pathways

inducing senescence. What they see is apoptosis and engulfment of unfit cells, nothing to do with the terminal fate of senescence. I think many of those conceptual problems need to be solved before it is a readable piece and the reader can make sense of their data. The many conceptual confusions the authors have will only confuse the reader and the field.

- The data that suggest that YAP acts different in healthy and injured hepatocytes, and could be a mechanism to eliminate damaged hepatocytes are interesting. It is however unclear whether this is a cell autonomous process or requires interactions with neighboring cells and hence comparison of cell fitness. It is possible that preexisting changes in cytoskeletal regulators in damaged cells modulate YAP function in a cell autonomous manner, without fitness comparison between cells. I think it is therefore also confusing to use the term unfit cells. It will be easier to just mention damaged cells.

We thank the reviewer for his/her insightful and valuable comments and agree to the suggestion.

First, we would like to explain the reason why we originally used the term “unfit cells”. Recently, we found that YAP-activating epithelial MDCK cells were extruded to the apical side in a non-cell autonomous manner and used the term “unfit cells” based on the concept of cell competition (revised reference #38). Next, we examined whether similar phenomenon occur in mouse liver by *in vivo* mosaic analysis shown here. As a result, we found a similar cellular response of YAP-activating hepatocyte elimination into sinusoids instead of hepatocyte proliferation. Therefore, we used the term “unfit cell” in the original MS.

However, as this reviewer points out, we could not show whether this cell response is cell autonomous or a non-cell autonomous process dependent on interactions with neighboring cells. So, we agree that the term “unfit cell” is inadequate at this stage.

Therefore, we have deleted “unfit cells” and, instead, use “damaged or injured cells” in the revised MS.

To explain the terms damaged and transformed cells, we now cite revised references #1 and #4.

- At this stage it is also unclear what are the physiological consequences of this mechanism. Although I understand this is not easy to asses, it would be good to mention this uncertainty in the discussion. How important is this mechanism to maintain organ function?

We thank the reviewer for this important comment.

Under physiologically normal conditions, the Hippo pathway is constitutively activated

and endogenous YAP rapidly becomes phosphorylated and thereby inactive in mouse livers (revised Fig. 1a inset and 1c; revised reference #30). Conversely, when the Hippo pathway is inactivated by stress, YAP immediately becomes unphosphorylated and translocates into the nucleus and induces target gene expression. So, we consider that the Hippo-YAP signal is used as a stress sensor to maintain tissue and organ homeostasis (revised Supplementary Fig. 11).

We have now added sentences in the “Discussion”.

“However, our study has shown that YAP activation specifically in damaged hepatocytes triggers their elimination in normal liver. The Hippo pathway is constitutively activated and rapidly inactivates YAP by phosphorylation. Conversely, when the Hippo pathway is inactivated by stress, YAP immediately becomes unphosphorylated, translocates into the nucleus and induces target gene expression. Based on this, it is considered that YAP plays a role in an emergency stress response to maintain tissue homeostasis due to the elimination of injured cells.”

REVIEWERS' COMMENTS:

Reviewer #1 (Remarks to the Author):

The authors satisfactorily answered most of the concerns raised by this reviewer, except points 4 and 5.

- The authors stated in the manuscript "Thus, specifically in the presence of both activated YAP and cellular damage induced by HTVi, hepatocyte fate changes from proliferation to migration/apoptosis" (see page 8 new manuscript). In Point #4, my question "Why only hydrodynamic injection and alcohol administration induce this YAP-induced hepatocyte fate to switch between proliferation to migration/apoptosis?" is not addressed.

The fact that different type of injuries affect different cell populations in the liver was already explained in the previous version of the manuscript and clear to me, but it does not answer the question.

- Point #5 was not addressed. The IF pictures shown in Fig. S5 of the revised manuscript are not conclusive and do not prove "hepatocyte migration".

Reviewer #2 (Remarks to the Author):

accept

Reviewer #3 (Remarks to the Author):

The MS is improved and ready to be published. Just I minor suggestion, not mandatory, I think it would make sense to mention the unfit cells in the intro with a citation to a recent review like: Merino MM, Levayer R, Moreno E. Survival of the Fittest: Essential Roles of Cell Competition in Development, Aging, and Cancer. Trends Cell Biol. 2016 Jun 16. pii: S0962-8924(16)30052-6. doi: 10.1016/j.tcb.2016.05.009.

Just for the sake of clarifying concepts.

Major Changes to Manuscript ID#: NCOMMS-16-23652-T and Specific Responses to Reviewers' Comments

I. Specific Responses to Reviewer #1:

"The authors satisfactorily answered most of the concerns raised by this reviewer, except points 4 and 5."

We thank the reviewer for this comment.

- The authors stated in the manuscript "Thus, specifically in the presence of both activated YAP and cellular damage induced by HTVi, hepatocyte fate changes from proliferation to migration/apoptosis" (see page 8 new manuscript). In Point #4, my question "Why only hydrodynamic injection and alcohol administration induce this YAP-induced hepatocyte fate to switch between proliferation to migration/apoptosis?" is not addressed. The fact that different type of injuries affect different cell populations in the liver was already explained in the previous version of the manuscript and clear to me, but it does not answer the question.

As you know, the liver has a unique dual blood supply mediated by the hepatic artery and the portal vein. Both of these subdivide within the liver into small capillaries called liver sinusoids. These sinusoids are made up of liver sinusoidal endothelial cells (LSECs), which cover the hepatic cords. Thus, hepatocytes are covered and protected by LSECs. We found that only injury to **both** LSECs and hepatocytes, which occurred only by hydrodynamic injection or by ethanol, is required for significant hepatocyte elimination (see Table below). Hepatocyte injury may attenuate cell-cell adhesion, and LSEC injury may be induced when breaking the physical barrier that covers hepatocytes. We think it is reasonable to propose that only injured hepatocytes that have also lost adhesion with neighbouring LSECs, induce the activation of YAP and thereby move to the sinusoids. Without the co-occurring LSEC damage, the hepatocyte would be unable to migrate, and instead undergo YAP-mediated proliferation, as shown by Su et al. (eLife 2015: ref 39). While this scenario is interesting to consider, it is still speculation at this stage. We could add an additional explanation to the manuscript if necessary.

Figure

Table. Different type of injuries affect different cell populations

	Injury		Cell elimination	Reference
	Hepatocytes	LSECs		
HTVi	+	+	+	The AAPS Journal 2010
CCl ₄	+	—	—	Toxicol Appl Pharmacol. 2012: Ref 25
Monocrotaline	—	+	—	Hepatology 2006: Ref 26
EtOH	+	+	+	J Dig Dis 2011: Ref 28

- Point #5 was not addressed. The IF pictures shown in Fig. S5 of the revised manuscript are not conclusive and do not prove “hepatocyte migration”. "

We presented new data that more clearly showed that YAP (wild-type)-expressing hepatocytes located to hepatic cords (revised Supplementary Fig. 5). In contrast, active YAP-expressing hepatocytes were surrounded by Stab2 or LYVE1-positive LSECs. From these results, we concluded that active YAP -expressing hepatocytes migrated to the sinusoid. I am afraid we cannot think of another plausible explanation for the appearance of these cells in the sinusoid other than that they have migrated there. In further support, we showed Cdc42-related genes are upregulated by activated YAP (revised Supp Table1) and identified Ect2 and Fgd3, which are guanine nucleotide exchange factors (GEF) for CDC42 and Rac (revised Figure 4d, e), both of which play central roles in cell migration. Indeed, Cdc42 or Rac1 dominant negative mutants

impaired hepatocyte elimination (which we conclude to be movement to the sinusoid) (revised Figure 4c). Together, we described that active YAP-expressing hepatocytes migrated to hepatic sinusoids, “sinusoid intra-vasation of hepatocytes” in the revised manuscript, which we think is a very reasonable conclusion given the data presented. This reviewer appears to disagree, but the alternative experiments are unfortunately technically impossible. Hepatocytes are highly polarized epithelial cells and form cords. Their basolateral surfaces face fenestrated LSECs, facilitating the exchange of materials between hepatocytes and blood vessels. Tight junctions formed between hepatocytes create a canaliculus that surrounds each hepatocyte. Bile secreted from mature hepatocytes is exported sequentially through bile canaliculi surrounded by the apical membrane of neighboring hepatocytes. Therefore, reconstruction of hepatic cords and LSECs by co-culture using cell lines or primary cells *in vitro* to show migration more directly is not currently possible. We could add a sentence acknowledging this limitation in the discussion if necessary.

II. Specific Responses to Reviewer #2:

accept

We appreciate the reviewer’s decision.

III. Specific Responses to Reviewer #3:

The MS is improved and ready to be published. Just I minor suggestion, not mandatory. I think it would make sense to mention the unfit cells in the intro with a citation to a recent review like:

Merino MM, Levayer R, Moreno E. Survival of the Fittest: Essential Roles of Cell Competition in Development, Aging, and Cancer. Trends Cell Biol. 2016 Jun 16. pii: S0962-8924(16)30052-6. doi: 10.1016/j.tcb.2016.05.009.

Just for the sake of clarifying concepts.

We thank the reviewer for his/her comments.

We added the paper as reference #9.